# OpenReview forum: "Rethinking LLM Ensembling from the Perspective of Mixture Models"
_ICML.cc/2026/Conference — ICML 2026 spotlight_

### Official Review · Reviewer_9fsu · 2026-03-11

**Soundness:** 2
**Presentation:** 3
**Significance:** 2
**Originality:** 2
**Overall Recommendation:** 3
**Confidence:** 4

**Summary:**

This work frames LLM ensembling as sampling from a mixture model. Instead of running all n models to compute the averaged distribution, you randomly pick one model per token and sample from it. The math is a direct application of the law of total probability. A lazy KV cache sync strategy handles the practical challenge of model switching. Experiments on those benchmarks show ME matches conventional ensemble accuracy while running roughly 2~ faster.

**Compliance With Llm Reviewing Policy:**

Affirmed.

**Key Questions For Authors:**

(1) How does ME perform as sampling temperature decreases ? The equivalence holds exactly only under stochastic sampling. Quantifying the degradation curve would make the practical scope much clearer.

(2) Have you tested on models above 10B? Lazy KV cache sync requires keeping n separate caches in memory. At 70B this could become a real bottleneck.

**Limitations:**

The greedy decoding limitation is acknowledged in sec. 5. The scope of this restriction is understated. The paper also does not discuss the memory overhead of maintaining multiple KV caches at scale.

**Strengths And Weaknesses:**

### Strengths

(1) A clean, actionable insight that people have overlooked. The mixture model equivalence is elementary, but applying it to LLM ensemble and building the lazy KV cache mechanism around it turns a textbook observation into a practical speedup. The prefill phase trick for cache sync is well motivated by the memory-bandwidth-bound nature of decoding.

(2) Broad experimental configurations. table. 2 and 3 cover similar models, heterogeneous models, and mixed-size ensembles. fig. 3 tests across GPUs. ME consistently matches CE accuracy while approaching single model speed. The ablation on lambda (fig. 4) is a nice addition.

(3) Sec. 3.5 offers a good unifying lens: ensemble as the simplest token level routing. Tab. 1 lays out the cost vs. performance tradeoff cleanly. This good angle gives us a useful mental model for situating ME among related approaches.

### Weaknesses

(1) Baseline coverage has a serious gap. ME is only compared against vanilla CE (sequential and parallel). DeePEn [1] is cited in sec. 2 but absent from experiments. It targets the same problem. CITER [2] is also a natural token level routing baseline that would ground the conceptual connection in sec. 3.5 with actual numbers. LLM Blender [3] is also missing. Right now the experiments only show ME beats the weakest possible baseline.

(2) Eq. 2 is the law of total probability. The novelty is slim. On top of that, the equivalence breaks down for greedy decoding and beam search (sec. 5), which are standard in code generation and structured output tasks. No experiment measures how ME degrades as temperature drops toward zero.

(3) Model scale is too small the icml venue. Every experiment sits in the 1.5 -8B range. It is unclear whether lazy KV cache sync remains cheap at 70B, where KV caches consume a large fraction of GPU memory.


- [1] Huang et al., Ensemble Learning for Heterogeneous Large Language Models with Deep Parallel Collaboration. NeurIPS24.
- [2] Zheng et al., CITER: Collaborative Inference for Efficient LLM Decoding with Token-Level Routing.
- [3] Jiang, Ren & Lin. LLM-Blender: Ensembling Large Language Models with Pairwise Ranking and Generative Fusion, ACL23.

---

> ### Author Rebuttal · Authors · 2026-03-31
>
> We sincerely thank the reviewer for their insightful and constructive feedback. We address your concerns below:
>
> **W1: Comparison with DeePEn, CITER, and LLM-Blender**
>
> We thank the reviewer for pointing out these works. However, comparing ME against these baselines fundamentally misaligns with our problem setting and core contribution.
>
> Our goal is **not** to propose a new state-of-the-art ensemble method to maximize absolute performance. Instead, our core contribution is proving that CE can be executed at a fraction of the computational cost while maintaining strict mathematical equivalence.
>
> - **DeePEn and LLM-Blender:** Both methods require explicit forward passes from *all* constituent models at every decoding step to compute latent representations or perform generative fusion. Their inference cost is inherently $O(N)$. In contrast, ME achieves an $O(1)$ inference cost. Comparing their absolute performance against ME without accounting for this massive efficiency gap would overlook the fundamental difference in their computational paradigms.
> - **CITER:** This is a *training-based* token-level routing method designed to route easy tokens to small models and hard tokens to large models. ME, conversely, is a **training-free, plug-and-play** mechanism that strictly mathematically recovers CE.
>
> Finally, we must clarify the statement that "ME beats the weakest possible baseline." **ME does not aim to "beat" CE.** Our fundamental claim is exactly that ME *matches* CE's performance perfectly, but does so several times faster. This mathematically equivalent acceleration is exactly the value of our work.
>
> **W2 & Q1: Theoretical Novelty and Temperature Degradation**
>
> We agree that the underlying property is a standard mathematical concept. However, our work offers a novel reinterpretation within the context of LLM ensembling, rendered practical by our system-level optimization (lazy KV-cache synchronization). While the math is straightforward, its application is non-trivial; if this equivalence were obvious, prior works would not have continued the computationally expensive practice of evaluating all models at every single step.
>
> Regarding performance degradation as temperature drops (Q1): We would like to clarify that because our method is a mathematically exact reformulation, the equivalence between ME and CE remains strictly preserved across any sampling distribution. Consequently, ME does not experience a degradation curve relative to CE as the temperature decreases. It perfectly matches CE's output regardless of how low the temperature is, provided the decoding relies on sampling rather than a deterministic argmax.
>
> Regarding greedy decoding (W2): While greedy decoding was traditionally standard, the industry paradigm has rapidly shifted. With the rise of reasoning models, sampling-based generation is now heavily favored even for highly deterministic structured tasks like math and coding. For instance, the official guidelines for state-of-the-art reasoning models—including DeepSeek-R1 [1], Kimi K2.5 [2], and QwQ [3]—explicitly mandate sampling with high temperatures (e.g., T=0.6 to 1.0) rather than greedy decoding. Therefore, ME remains highly relevant and practical for modern LLM deployment.
>
> **W3 & Q2: Model Scale and KV Cache Memory**
>
> We understand the reviewer's desire to see evaluations on larger models. However, we would like to highlight that our core contribution lies in establishing a fundamental mathematical and empirical equivalence. Because this equivalence is mathematically agnostic to model architecture or parameter count, we believe our robust experiments across the 1.5B-8B range effectively validate this core claim.
>
> While evaluating an ensemble of 70B models exceeds our current academic computational resources, the theoretical guarantees and latency reductions of ME naturally scale to larger models. Regarding memory overhead (Q2), we acknowledge that maintaining multiple KV caches in memory is an inherent requirement for *any* ensemble method. ME's primary breakthrough is not memory reduction, but rather eliminating the redundant **compute (FLOPs) and memory bandwidth** bottlenecks during the generation phase, which are the true limiting factors for ensemble latency.
>
>
>
> [1]. DeepSeek-R1: Incentivizing Reasoning Capability in LLMs via Reinforcement Learning.
>
> [2]. Kimi K2.5: Visual Agentic Intelligence.
>
> [3]. QwQ: Reflect Deeply on the Boundaries of the Unknown.

---

### Official Review · Reviewer_1Bzu · 2026-03-12

**Soundness:** 3
**Presentation:** 3
**Significance:** 2
**Originality:** 3
**Overall Recommendation:** 5
**Confidence:** 3

**Summary:**

This work introduces a cost-effective, mixture-model-like ensemble methodology for autoregressive language models. Instead of defining the next-token-prediction distribution as a mixture of categorical distributions from all ensemble members, the proposed approach employs a stochastic strategy; it utilizes the categorical distribution of a randomly selected ensemble member at each step of the left-to-right generation process. Empirical results cover a range of open-source models (Llama, Qwen, Mistral, etc.) with scales of up to 8B parameters.

**Compliance With Llm Reviewing Policy:**

Affirmed.

**Final Justification:**

I appreciate the active participation of both the authors and the committee. Consistent with my initial rebuttal acknowledgement, I raised my score with the expectation that the clarifications and discussions provided during this period will be thoroughly integrated into the final manuscript.

**Key Questions For Authors:**

- Conceptually, as the authors discussed, the proposed ME strategy can be viewed as a token-level routing mechanism employing a fixed random router. Have the authors compared this stochastic approach against a version where the router is explicitly trained?
- This may be a matter of stylistic preference, but I suggest moving Table 1, Algorithm 1, and Algorithm 2 to the top of their respective pages. In the current version, the main text is sandwiched between these floating elements, which made me lose my place while reading.

**Limitations:**

yes

**Strengths And Weaknesses:**

Strengths.
- The proposed methodology offers a practical, cost-effective solution to the challenges of ensembling LLMs. The discussion of realistic scenarios in Sections 3.3 and 3.4 is a particularly strong point, grounded in practical considerations.
- The experimental design seems to be robust. It provides a comprehensive evaluation across various dimensions, including standard benchmarks as well as practical throughput gains, well-conducted on a diverse range of models up to the 8B-parameter scale.
- Overall, the manuscript is well-organized and easy to follow. The empirical findings (with tables and figures) are also presented clearly.

Weaknesses.
- I hate to be the reviewer who asks for more compute, but for an empirical paper in this field, demonstrating that the proposed strategy holds for larger-scale models would strengthen the contribution.
- Another factor behind the decline of ensembling in the era of large models is that an aggregate of weak models often fails to match the performance of a single model of comparable total scale. For instance, an ensemble of two or three 7B models typically yields results inferior to those of a single 14B or 21B model. The proposed ME strategy still inherits this fundamental limitation, much as in CE.

---

> ### Author Rebuttal · Authors · 2026-03-31
>
> We sincerely thank the reviewer for their insightful and constructive feedback. We address your concerns below:
>
> **W1: Scaling to Larger Models**
>
> We deeply appreciate your empathy regarding compute limitations. You are completely right that testing on larger-scale models would ideally strengthen an empirical paper. However, we would like to candidly note that our primary focus is proving the strict mathematical and empirical **equivalence** between ME and CE, rather than establishing SOTA absolute performance.
>
> Due to resource constraints during the short rebuttal period, we are unfortunately unable to scale our ensemble tests to larger models. Nevertheless, because ME is fundamentally a mathematical reformulation of CE, we are confident that this exact equivalence holds true regardless of the underlying model scale.
>
> **W2: Limitation of Ensembling vs. Larger Models**
>
> We completely agree with your premise. An ensemble of two 7B models typically underperforms a single 14B model. However, we believe ME actually alters the practical speed-performance trade-off of this comparison.
>
> We believe that historically, the decline of ensembling was largely driven by inference latency: ensembling two 7B models requires two forward passes per decoding step, making it significantly slower and more compute-intensive than querying a single 14B model. Under these conditions, conventional ensembling was often not worth the delay.
>
> ME fundamentally changes this dynamic. With ME, ensembling two 7B models runs at the inference speed of **a single 7B model**. Therefore, even if an ME of two 7B models cannot beat a 14B model, it only needs to outperform a single 7B model to be highly valuable—offering a performance boost essentially "for free" in terms of generation latency. Furthermore, in practical scenarios where a 14B model is simply unavailable, ME provides a highly efficient way to squeeze out extra performance from existing weaker models.
>
> **Q1: Comparing ME with Trained Routers**
>
> To be perfectly candid: no, we have not compared ME against explicitly trained token-level routers. As you correctly intuited, it is highly unlikely that a fixed stochastic router (ME) could outperform a router explicitly optimized and trained on domain-specific data.
>
> Our objective in this paper is not to propose ME as a SOTA routing algorithm to beat trained routers. Instead, our goal is to present it as a mathematically equivalent, high-speed substitute for CE. Compared to routing methods, the unique advantage of ME (and CE) is that it is strictly **plug-and-play and training-free**. It entirely bypasses the need for the data collection, training time, and compute overhead required to train a dedicated router.
>
> **Q2: Formatting Adjustments**
>
> Thank you so much for pointing this out! We completely agree that the current placement disrupts the reading flow. We will certainly move Table 1, Algorithm 1, and Algorithm 2 to the top of their respective pages in the revised manuscript to improve readability.

---

> > ### Author Rebuttal · Reviewer_1Bzu · 2026-04-01
> >
> > W1. I fully understand the context regarding this point. The authors may rest assured that I no longer consider this an issue, and I will exclude it from any further discussion or assessment.
> >
> > W2. Focusing on the latency-performance trade-off rather than memory-performance is a strategic positioning, but I have reservations about its practicality. The ME strategy implies a use case where memory is abundant but speed is critical, allowing a slight performance boost by loading extra models without a significant latency hit. However, given that HBM capacity is typically the most constrained resource in modern LLM inference, it remains unclear whether a 'surplus memory' scenario is common enough to make this approach a realistic solution for most use cases.
> >
> > Q1. I acknowledge the simplicity of the proposed training-free strategy, but it feels like a missed opportunity not to demonstrate how it stacks up against the performance levels of an optimized router. Building on my previous point: for a user who has surplus memory but opts for a smaller model to maintain speed, it is highly likely they would be willing to incur some training costs for an optimized router if it yields superior performance.
> >
> > Q2. I appreciate the authors for adopting my suggestion.
> >
> > Overall, while some practical concerns remain, I recognize the empirical validation of the ME approach matching CE performance as a key contribution. The authors' response to Reviewer DbUu confirms its statistical rigor, and they appear committed to making the empirical validation more comprehensive by addressing Reviewer UL96's constructive feedback. Expecting the full results to be included in the final manuscript, I am raising my score.

---

### Official Review · Reviewer_UL96 · 2026-03-13

**Soundness:** 3
**Presentation:** 4
**Significance:** 3
**Originality:** 3
**Overall Recommendation:** 5
**Confidence:** 3

**Summary:**

Conventional ensembling models (CE) usually require making a forward pass for every model step and then computing the average of the output probability for every token generation step. This work presents Mixture-model-like ensembling (ME), proposing that ensembling is equivalent to stochastically sampling a single model at each token generation step. However, in ME, sampling a single model still requires full forward passes for each model due to kv caching (i.e. if we selected another models for previous steps but switch to a different model for the current step, we still need to recompute the kv values on previous tokens for the current model). Authors propose lazy synchronization, which prefills previous tokens for a model only if that model is selected. This is still efficient since inference is usually memory-bound.

On several benchmarks, authors show CE and ME achieve similar performance, while the proposed ME has better efficiency.

Authors also show that one can ensemble models of different sizes with ME to have a trade-off between performance and speed, controlled by hyperparameter $\lambda$.

**Compliance With Llm Reviewing Policy:**

Affirmed.

**Final Justification:**

The rebuttal addressed my main concerns, hence I'm increasing my score.

**Key Questions For Authors:**

See weakness.

**Limitations:**

Yes.

**Strengths And Weaknesses:**

Strengths

- The problem of improving ensembling efficiency is well motivated

- The paper's writing and figures is clear and easy to understand

- The proposed method is simple and effective solution, demonstrating clear gains in efficiency while maintaining performance.



Weaknesses

- The paper has a very limited evaluation, only on 4 tasks, all on the simpler side. I would also like to see more and harder tasks to demonstrate that the method is generalizable on all benchmarks, such as:

  - Instruction following: IFEval, Arena-Hard V2
  - Knowledge: MMLU-Pro in addition to MMLU
  - Math: MATH in addition to GSM8K
  - Coding: MBPP and HumanEval
  - Science: GPQA

- From a performance perspective, I think ensembling or token routing is usually most effective when the models are each good at different areas. I see in the paper that authors have attempted to do this by including Qwem Math, can authors also include other specialized models such as Qwen Code?

- Given that the methods and baselines require no training, the paper should have been evaluated on larger models i.e., the larger Qwen and deepseek models. I’m curious to see whether larger models, which tend to saturate on some benchmarks, will still see improvements with ensembling.

- For ensembling models with a heterogeneous vocabulary set, does it require small amounts of training since it needs to learn the function mapping for vocab alignment? If yes, authors then also need to compare to token routing baselines

---

> ### Author Rebuttal · Authors · 2026-03-31
>
> We sincerely thank the reviewer for their insightful and constructive feedback. We address your concerns below:
>
> **W1, W2 & W3: More Tasks, Specialized Models, and Larger Scale**
>
> We thank the reviewer for the constructive suggestions. While evaluating larger models on harder tasks indeed demonstrates the absolute performance upper bound of ensembling, we would like to politely emphasize that our core contribution is establishing strict mathematical and empirical equivalence between ME and CE, rather than proposing specific model combinations to achieve SOTA results. As long as ME consistently and accurately replicates CE's performance at a fraction of the computational cost, our core claim holds true, independent of specific downstream datasets or absolute model scale.
>
> Nevertheless, to directly address your interest in specialized models and harder tasks, we prioritized the following representative evaluations during the limited rebuttal period:
>
> - **Math:** Qwen-3B + Qwen-1.5B-Math on the MATH dataset.
> - **Coding:** Qwen-3B + Qwen-Coder-3B on HumanEval.
>
> Detailed results are provided here: [link](https://anonymous.4open.science/r/ME-Rebuttal-Supplement-ICML26-B2D3/qwen_ensemble_math_humaneval_eval.md). The results clearly demonstrate that even in these specialized domains, ME perfectly matches CE's performance while maintaining significant inference acceleration. While comprehensively evaluating larger models across massive, high-difficulty benchmarks is computationally prohibitive during the short rebuttal phase, our theoretical equivalence mathematically guarantees that this efficiency and performance consistency will naturally generalize to larger-scale models.
>
> **W4: Vocabulary Alignment and Training Baselines**
>
> We appreciate the opportunity to clarify this. First, to directly answer your question: no, the vocabulary alignment method (UniTe) we used requires absolutely no training; it relies entirely on pre-computed heuristic mappings.
>
> More importantly, to see why ME does not need to be compared to token-routing baselines, we must decouple the heterogeneous decoding pipeline into two distinct stages:
>
> 1. **Stage 1: Vocabulary Alignment.** This stage maps heterogeneous vocabularies into a shared space. While some methods use training here and others (like UniTe) are training-free, this stage is completely **orthogonal** to our contribution.
> 2. **Stage 2: Model Aggregation / Selection.** This is where ME operates. In CE, aggregating the aligned models requires *no training*, but is extremely slow because it evaluates all models. In token-level routing, selecting a model requires a *training-based* router.
>
> ME belongs strictly to the ensembling family at Stage 2: it is a **training-free** selection mechanism that perfectly replicates CE's aggregation result but only requires evaluating a single model. Because ME inherently does not involve any learned routing parameters to boost absolute performance, it is structurally different from, and orthogonal to, training-based routing baselines.

---

> > ### Author Rebuttal · Reviewer_UL96 · 2026-04-04
> >
> > Dear reviewers,
> >
> > Thank you for your carefully written rebuttal
> >
> > Regarding W1, W2, W3: I appreciate the authors taking the time to run the extra results, and I'm satisfied to see that on specialized models, ME matches CE performance. I also understand that authors are compute-limited to run additional evaluations on larger models.
> >
> > Regarding W4: Thank you for the clarifications on prior work.
> >
> > I'm overall happy with the rebuttal, so I'm increasing my score.

---

### Official Review · Reviewer_DbUu · 2026-03-13

**Soundness:** 3
**Presentation:** 3
**Significance:** 3
**Originality:** 3
**Overall Recommendation:** 5
**Confidence:** 4

**Summary:**

The paper investigates whether stochastic LLM ensembling requires evaluating all models at every decoding step. It shows that ensemble sampling can be reformulated as mixture sampling: instead of averaging next-token distributions from all models, one can sample a model according to the ensemble weights and then sample a token from that model, yielding the same next-token distribution under the sampling regime. The paper also proposes a lazy KV-cache synchronization mechanism to make this approach efficient in practice, relates the method conceptually to token-level routing, and reports substantial inference speedups over conventional ensembling while maintaining comparable empirical accuracy

**Compliance With Llm Reviewing Policy:**

Affirmed.

**Final Justification:**

The authors addressed most of the questions and provided the necessary statistical analysis

**Key Questions For Authors:**

1. **Broader methodological implications of the ME perspective**
  Can the authors clarify the broader methodological implications of the Mixture-model-like Ensemble formulation? If simple token-level routing can recover classical stochastic ensembling, does this perspective enable a more systematic transfer of ideas from classical ensemble learning to multi-LLM routing or collaboration methods? A clearer discussion of what this viewpoint unlocks beyond the specific ME construction would help assess the conceptual significance of the work

2. **Rationale behind the experimental design**
 The experiments are grouped into categories such as “similar”, “heterogeneous”, and “different sizes” but these categories combine multiple factors simultaneously (model family, tokenizer/vocabulary, architecture, specialization, and scale). Could the authors clarify the rationale behind the chosen model combinations and whether they considered a more controlled experimental matrix that isolates these factors more clean?

3. **Statistical support for accuracy comparisons**
  The paper reports five runs for accuracy and argues that ME and CE achieve comparable performance, but the main tables do not provide confidence intervals or statistical tests for these comparisons. Given that the central claim suggests equivalence between ME and conventional ensembling, could the authors provide uncertainty estimates or statistical tests demonstrating that the observed differences are statistically indistinguishable?

4. **Sensitivity to vocabulary alignment methods**
  For heterogeneous-model ensembles, tokenizer and vocabulary alignment plays an important role. The experiments use UniTe with top-k ensembling, but it is unclear how sensitive the results are to the alignment method. Could the authors comment on how robust the conclusions are to different alignment procedures, and whether the claimed equivalence remains meaningful once alignment approximations are considered?

5.**Intended real-world use cases**
  The mixture-model equivalence relies on sampling-based decoding and does not hold under greedy decoding. Could the authors elaborate on the intended deployment scenarios for ME, and identify the practical settings where the proposed approach is expected to provide the most benefit?

**Limitations:**

The paper briefly discusses limitations, noting that the proposed equivalence relies on sampling-based decoding and does not apply to greedy decoding. However, the discussion of limitations could be expanded. In particular, it would be helpful to comment more explicitly on the practical scope of the method (e.g., scenarios where sampling-based decoding is commonly used), as well as on potential limitations related to vocabulary alignment and the overhead of KV-cache synchronization. The societal impact of the work is likely limited, as the paper focuses on algorithmic aspects of LLM inference

**Strengths And Weaknesses:**

### Strengths

- **Clean theoretical insight**
  The paper presents a simple and elegant reformulation of sampling-based LLM ensembling. The derivation is straightforward and, under the stated assumptions, the proposed procedure is not merely heuristic but an exact reformulation of the ensemble distribution in the sampling regime.

- **Meaningful systems perspective**
  The proposed lazy KV-cache synchronization mechanism provides a practical way to avoid recomputing full prefixes when switching between models. The reported speed improvements across multiple hardware platforms (H100, RTX 3090, V100, A100) strengthen the practical relevance of the work.

- **Conceptual link to routing**
  Framing ensembling as a minimal form of token-level routing offers an interesting perspective on multi-model collaboration and may inspire future work on lightweight routing strategies.

### Weaknesses

- **Limited theoretical novelty**
  The core theoretical observation (that sampling from a mixture distribution can be implemented by first sampling a component and then sampling from that component) is a well-known property of mixture models. While the paper successfully applies this idea to LLM ensembling and develops a practical implementation around it, the underlying theoretical insight itself appears relatively straightforward. The paper could more clearly position its contribution as a practical reinterpretation and system-level optimization rather than a fundamentally new theoretical framework

- **Experimental design is not systematically controlled**
  The evaluation groups model combinations into categories such as “similar,” “heterogeneous,” and “different sizes,” but these categories mix multiple factors simultaneously (model family, architecture, tokenizer differences, and scale). As a result, it is difficult to isolate which properties actually drive the observed behavior. The selection of model combinations appears somewhat ad hoc rather than designed to test clear hypotheses

- **Accuracy comparisons lack statistical rigor**
  While statistical significance is reported for speed improvements, the same level of rigor is not applied to accuracy comparisons. The results show small fluctuations between the proposed method and conventional ensembling, sometimes slightly higher and sometimes slightly lower. Without confidence intervals or formal statistical testing, it is difficult to determine whether these differences reflect evaluation noise or meaningful deviations

- **Novelty relative to token-level routing is somewhat unclear**
  Conceptually, the method resembles a simplified form of token-level routing where the router samples models according to fixed mixture weights. While the reinterpretation of ensembling through the lens of routing is interesting, the paper could better clarify what is fundamentally new beyond this perspective

- **Vocabulary alignment is underexplored**
  For heterogeneous-model ensembles, tokenizer and vocabulary mismatch is a central practical issue. The paper treats alignment largely as a plug-in component by adopting an existing alignment method. However, the sensitivity of the results to the alignment strategy and the potential approximation errors introduced by alignment are not analyzed

- **Limited practical scope**
  The theoretical equivalence relies on sampling-based decoding and does not extend to greedy or near-deterministic decoding regimes. Since many real-world deployment settings rely on such decoding strategies, the paper would benefit from a clearer discussion of when the proposed method is most practically useful

---

> ### Author Rebuttal · Authors · 2026-03-31
>
> We thank the reviewer for their constructive feedback:
>
> **W1: Theoretical Novelty**
>
> We agree that the mixture-model property is a well-known mathematical concept. However, our work offers a novel reinterpretation of this concept within the context of LLM ensembling, rendered efficient by our lazy KV-cache synchronization mechanism.
>
> While the underlying math is straightforward, we believe its application here remains non-trivial. If this equivalence were obvious in the context of LLMs, prior works would not have continued the computationally expensive practice of evaluating all models at every decoding step.
>
> We appreciate the feedback and will reframe our contribution as a practical reinterpretation and system-level optimization.
>
> **W2 & Q2: Experimental Design**
>
> We agree our evaluation is not a strictly controlled matrix isolating individual factors. Instead, our objective was to validate ME's efficiency and performance equivalence across **practically meaningful ensembling scenarios**. As noted in UniTe, ensembling is only practically useful when the base models are reasonably competitive. Mechanically forcing a controlled matrix may produce combinations that underperform the strongest single model, rendering the ME vs. CE comparison uninformative. Thus, our selection was intentionally scenario-driven:
>
> - **Similar Models:** To verify ME reliably reproduces CE in baseline settings.
> - **Heterogeneous Models:** To avoid arbitrary pairings, we directly adopted UniTe's configuration.
> - **Different Sizes:** We initially used Llama to broaden model diversity. To confirm generalizability, we have now evaluated a different-sized Qwen pair ([link](https://anonymous.4open.science/r/ME-Rebuttal-Supplement-ICML26-B2D3/qwen_ensemble_lambda_ablation.md)); these consistent results confirm our findings hold across model families.
>
> **W3 & Q3: Statistical Rigor of Accuracy**
>
> To address this, we have conducted 20 independent runs to compute the mean, std, and p-values. As detailed in [link](https://anonymous.4open.science/r/ME-Rebuttal-Supplement-ICML26-B2D3/stat_analysis_of_ce_and_me.md), the results yield $p > 0.05$ across all datasets, rigorously confirming the equivalence of ME and CE.
>
> **W4: Novelty vs. Token-Level Routing**
>
> Beyond a conceptual link, our perspective offers actionable utility for both perspectives:
>
> - **For Ensembling:** It breaks the historical bottleneck of explicitly evaluating all models, making stochastic ensembling viable at a fraction of the cost while maintaining exact mathematical equivalence.
> - **For Routing:** It elevates the "random router"—typically dismissed as a trivial baseline—into a powerful, **training-free routing method** mathematically guaranteed to match full-ensemble performance without any training overhead.
>
> **W5 & Q4: Vocabulary Alignment**
>
> To clarify, the mathematical equivalence between ME and CE is strictly preserved regardless of the vocabulary alignment method used.
>
> Because both ME and CE operate on the probability space after alignment is applied, any approximation errors or information loss introduced by the alignment process will affect both methods identically. Our core claim is simply that whatever aligned distribution CE produces, ME will faithfully reproduce it at a fraction of the cost.
>
>  **W6 & Q5: Real-World Use Cases**
>
> We acknowledge the greedy decoding limitation. However, with the recent industry shift towards **reasoning models**, sampling-based generation is rapidly replacing greedy decoding as the dominant paradigm.
>
> ME is specifically designed for these high-volume settings where generation diversity is essential and compute is a massive bottleneck:
>
> - **Test-Time Scaling:** Best-of-N and tree-search algorithms, which are foundational to modern reasoning models.
> - **Synthetic Data & RL:** Generating massive, diverse rollouts for alignment and self-improvement pipelines.
> - **Open-Ended Chat:** Standard user-facing applications operating at temperature > 0.
>
> **Q1: Broader Methodological Implications**
>
> We appreciate the reviewer's thought-provoking question. Interestingly, our perspective suggests the exact opposite: we argue that the historical inefficiency of LLM ensembling stems precisely from borrowing *too heavily* from classical ensemble learning.
>
> As noted in our Introduction, classical ML ensembles aggregate raw scores to select a single deterministic label (argmax). However, LLMs generate tokens via stochastic sampling. By explicitly decoupling LLM ensembling from classical ML paradigms, we reveal that they are fundamentally different operations.
>
> Therefore, rather than transferring more classical ML ideas, the broader methodological implication of our work lies in unlocking a new paradigm for **generalized, lossless LLM acceleration**. As detailed in **Appendix A.4**, our formulation allows us to probabilistically skip heavy computations for complex decoding strategies while strictly preserving the exact final output distribution.

---

> > ### Author Rebuttal · Reviewer_DbUu · 2026-04-04
> >
> > Thank you very much for the detailed written rebuttal!
> >
> > I appreciate the additional experimental effort and, overall, I am satisfied with the responses to my questions. One minor point is that the argument that *“equivalence were obvious in the context of LLMs, prior works would not have continued the computationally expensive practice of evaluating all models at every decoding step”* seems quite weak to me. However, in light of the clarifications provided for other responses, I consider this concern to be non-essential.
> >
> > Given the expected additions to the camera-ready version, I align with the other reviewers and raise my overall evaluation of the paper

---

### Decision · Program_Chairs · 2026-04-30

**Decision:**

Accept (spotlight)

**Comment:**

This paper presents a reformulation of sampling-based LLM ensembling as a mixture model. By selecting a single model at each decoding step, the proposed Mixture-model-like Ensemble (ME) achieves significant inference speedups while maintaining exact mathematical equivalent to conventional ensembling.

The application of mixture models to LLM ensembling addresses a major efficiency bottleneck. The introduction of a lazy KV-cache synchronization makes model switching efficient. The authors provide extensive analysis during the rebuttal to confirm that ME and CE are not distinguishable in accuracy across multiple benchmarks. The authors successfully argues that the sampling based decoding is increasingly the dominant paradigm for modern reasoning model.

Initially reviewers had concerns regarding model scale and limited baselines. The authors clarified that the establishment of a equivalence to CE, which is agnostic to scale and orthogonal to training-based routing methods. The paper is technically solid and offers immediate value for efficient LLM deployment.